# Direct Acquisition Optimization for Low-Budget Active Learning

**Zhuokai Zhao**[1]    **Yibo Jiang**[1]    **Yuxin Chen**[1]
[1]Department of Computer Science, University of Chicago,
Correspondence to zhuokai@uchicago.edu

## Abstract

Active Learning (AL) has gained prominence in integrating data-intensive machine learning models into domains with limited labeled data. However, its effectiveness diminishes significantly when the labeling budget is especially low. In this paper, we empirically verify the performance degradation of existing AL algorithms in the extremely low-budget settings, and then introduce *Direct Acquisition Optimization* (DAO), a novel AL algorithm that optimizes sample selections based on expected true loss reduction. Specifically, DAO utilizes influence functions to update model parameters and incorporates an additional acquisition strategy to mitigate bias in loss estimation. This approach facilitates efficient estimation of the overall error reduction, without extensive computations or reliance on labeled data. Experiments demonstrate the effectiveness of DAO in low-budget settings, outperforming state-of-the-arts approaches across seven benchmarks.

## 1 Introduction

Active learning (AL) explores how adaptive data collection can reduce the amount of data required by machine learning models, making it particularly valuable when labeled data is scarce or expensive [49, 46, 42, 59]. It is especially crucial for modern deep learning (DL) models as they are often data-hungry, and labeling can be cost-prohibitive [55]. AL algorithms strategically select the most beneficial data points for labeling, maximizing training efficiency even with limited data, and have been widely applied in fields such as medical imaging [7], astronomy [53], and physics [12]. In these cases, selecting samples for labeling can substantially reduce costs [42].

Over the years, various AL algorithms have emerged, from early contributions [31, 54, 44] to more recent approaches targeting deep learning models [21]. AL algorithms generally fall into two categories: heuristic-based objectives that differ from evaluation metrics, like diversity [47] and uncertainty [16], and methods that directly optimize evaluation metrics, such as expected error reduction (EER) [44] and its successors [24, 41]. Despite the popularity of heuristic methods, research and empirical analysis [39, 18] show these approaches often fail in low-budget settings, where less than 1% of data can be labeled. Algorithms that directly target error reduction, like EER, are computationally expensive and require retraining the model for each candidate, making them impractical for deep networks. Others, like GLISTER [24], require labeled validation sets, posing challenges in data-scarce cases where labeled data is too valuable to reserve for AL algorithms.

To address these limitations, we propose **Direct Acquisition Optimization (DAO)**, a novel AL algorithm that selects samples by efficiently estimating expected loss reduction, without relying on a labeled set. DAO overcomes the runtime challenges of methods like EER and GLISTER by leveraging influence functions [32] for model updates and employing a more efficient unbiased estimator of loss reduction through importance-weighted sampling. Our contributions include a novel, efficient AL algorithm that optimizes sample selection based on expected error reduction while

Workshop on Bayesian Decision-making and Uncertainty, 38th Conference on Neural Information Processing Systems (NeurIPS 2024).

avoiding labeled set dependencies. Extensive experiments show that DAO outperforms popular AL methods in low-budget scenarios across seven benchmarks.

## 2 Methodology

Different from the heuristics-based AL algorithms that optimize criteria such as diversity or uncertainty, DAO is built upon the EER formulation with the selection objective being the largest reduced error evaluated on the entire unlabeled set. More specifically, DAO majorly improves upon two aspects: (1) instead of re-training the classifier, we employ influence function [11], a concept with rich history in statistical learning, to formulate the new candidate sample as a small perturbation to the existing labeled set, so that the model parameters can be estimated without re-training; and (2) instead of reserving a separate, relatively large labeled set for validation [24], we sample a very small subset directly from the *unlabeled* set and estimate the loss reduction through bias correction.

Essentially, when considering each candidate from the unlabeled set, we optimize the EER framework on two of its core components, which are model parameter update and true loss

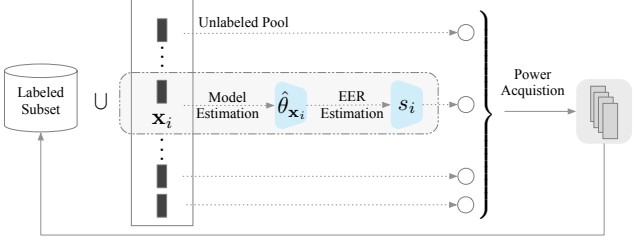

Figure 1: Schematic of the algorithmic framework of DAO.

estimation. Additionally, we upgrade EER, which only supports single sequential acquisition, to offer DAO in both single and batch acquisition variants by incorporating stochastic samplings to the sorted estimated loss reductions. We illustrate our algorithmic framework in Fig. 1.

### 2.1 Problem Statement

The optimal sequential active learning acquisition function can be formulated as selecting a budget number of samples $\mathbf{x}_t^{\text{train}}$ from the current unlabeled set $\mathcal{U}_t$ at each round $t$ such that

$$\mathbf{x}_t^{\text{train}} = \operatorname*{arg\,min}_{\mathbf{x}_{\mathcal{S}_i} \subset \mathcal{U}_{t-1}} \mathbb{E}_{(y_{\mathcal{S}_i}|f^*, \mathbf{x}_{\mathcal{S}_i})} \left[ L_{\text{true}}(f_{t|\mathbf{x}_{\mathcal{S}_i}, y_{\mathcal{S}_i}}) \right] \tag{1}$$

where $f^*$ represents an optimal oracle that maps from any subset of the unlabeled data $\mathbf{x}_{\mathcal{S}_i} \in \mathcal{U}_{t-1}$ to their ground-truth labels $y_{\mathcal{S}_i}$, and $f_{t|\mathbf{x}_{\mathcal{S}_i}, y_{\mathcal{S}_i}}$ is the model that has been trained on the union of the current labeled set $\mathcal{L}_{t-1}$ and the current unlabeled candidates $\mathbf{x}_{\mathcal{S}_i} \in \mathcal{U}_{t-1}$. In addition, $L_{\text{true}}(f_{t|\mathbf{x}_{\mathcal{S}_i}, y_{\mathcal{S}_i}}) = \frac{1}{|\mathcal{U}_{t-1,i}|} \sum_{\mathbf{x} \in \mathcal{U}_{t-1,i}} \ell(\mathbf{x}; f_{t|\mathbf{x}_{\mathcal{S}_i}, y_{\mathcal{S}_i}})$ represents the loss estimator that can predict the *unbiased* error of $f_{t|\mathbf{x}_{\mathcal{S}_i}, y_{\mathcal{S}_i}}$, where $\ell$ denotes the loss function. It is numerically the same as if $f_{t|\mathbf{x}_{\mathcal{S}_i}, y_{\mathcal{S}_i}}$ has been tested on the entire unlabeled set $\mathcal{U}_{t-1,i}$, where $\mathcal{U}_{t-1,i} = \mathcal{U}_{t-1} \setminus \{\mathbf{x}_{\mathcal{S}_i}\}$. Such formulation represents the optimal AL criterion and aligns with any existing sequential active learning algorithm — of which the goal is to select the new data points that can most significantly improve the current model performance [44]. Unfortunately, Eq. (1) cannot be directly implemented in practice. Because, first, we do not have access to the optimal oracle $f^*$ to reveal the labels $y_{\mathcal{S}_i}$ of $\mathbf{x}_{\mathcal{S}_i} \subset \mathcal{U}_{t-1}$; second, even if we had $f^*$ and therefore $y_{\mathcal{S}_i}$, we cannot afford the cost of retraining model $f_{t-1}$ on each $\mathcal{L}_{t-1} \cup \mathbf{x}_{\mathcal{S}_i}$ to obtain the updated $f_{t|\mathbf{x}_{\mathcal{S}_i}, y_{\mathcal{S}_i}}$; and third, we do not have the unbiased true loss estimator $L_{\text{true}}$, which demands evaluating $f_{t|\mathbf{x}_{\mathcal{S}_i}, y_{\mathcal{S}_i}}$ on the entire $\mathcal{U}_{t-1,i}$. Therefore, the goal of DAO is to solve the above challenges and efficiently and accurately approximate Eq. (1) for the sample selection strategy. It is also worth noting that, when $\mathbf{x}_t^{\text{train}}$ represents a *set* of newly acquired data points, the above formulation becomes eligible for batch active learning, which is more suitable for deep neural networks [21].

### 2.2 Label Approximation via Surrogate

In this section, we address the first challenge when approximating Eq. (1). As we do not know the true label or true label distribution $p(y|\mathbf{x}, f^*)$ of each unlabeled sample $\mathbf{x}$, the best we can do is provide an approximation for $p(y|\mathbf{x})$. To this end, we introduce the concept of a *surrogate* [27], which is a model parameterized by some potentially infinite set of parameters $\theta$. Specifically, $p(y|\mathbf{x})$ can be approximated using the marginal distribution $\pi(y|\mathbf{x}) = \mathbb{E}_{\pi(\theta)}[\pi(y|\mathbf{x}, \theta)]$ with some proposal

distribution $\pi(\theta)$ over model parameters $\theta$. In other words, we have:

$$p(y|\mathbf{x}) \approx \int_\theta \pi(\theta)\pi(y|\mathbf{x},\theta)\,\mathrm{d}\theta \tag{2}$$

As the sample selection process continues, new labeled points should also be used to train and update the surrogate model $\pi(\theta)$ for better approximation of the true outcomes.

Although ideally, a more capable surrogate is preferred for better ground truth approximations, we acknowledge that the choice of surrogate model can be very sensitive to the computational constraints. Therefore, if running time is at center of the concerns during sample acquisitions, using $f_t$ at step $t$ also as the surrogate could be an efficient alternative, as we don't need to update a second model, nor do we need to run forward pass on the both models. However, this will come with the cost that $\pi_t$ never disagrees with $f_t$, which causes performance degradation for the unbiased true loss estimation, which will be illustrated with more details in §2.4. Therefore, in short, we do not recommend replicating $f_t$ as surrogate in practice, unless the computational constraint is substantial.

### 2.3 Model Parameters Update without Re-training

At acquisition round $t$, suppose we have labeled set $\mathcal{L}_{t-1}$ and unlabeled set $\mathcal{U}_{t-1}$ as the results from the previous round $t-1$, and new sample $\mathbf{x}_i \in \mathcal{U}_{t-1}$ that is currently under consideration for acquisition, the goal of this section is to estimate the parameters of model $f_{t|\mathbf{x}_i,y_i}$ that could has been obtained after training $f_{t-1}$ on the combined dataset $\{\mathcal{L}_{t-1} \cup \mathbf{x}_i\}$. Here, $y_i$ denotes the (unobserved) ground-truth label of $\mathbf{x}_i$. In other words, if we suppose the conventional full training converges to parameters $\hat{\theta}_{\mathbf{x}_i}$, we have:

$$\hat{\theta}_{\mathbf{x}_i} = \arg\min_{\theta \in \Theta} \frac{1}{|\mathcal{L}_{t-1}| + 1} \sum_{\mathbf{x} \in \{\mathcal{L}_{t-1} \cup \mathbf{x}_i\}} \ell(\mathbf{x}; \theta) \tag{3}$$

where recall that $\ell(\mathbf{x}; \theta)$ denotes the loss of $\theta$ on $\mathbf{x}$. This objective is infeasible to compute exactly as $y_i$ is unknown and retraining is expensive even if $y_i$ is given. The core of our approach is that, instead of re-training as showed in Eq. (3), we can approximate the effect of adding a new sample as upweighting the influence function by $\frac{1}{|\mathcal{L}_{t-1}|+1}$ [26] and then directly estimate the updated model parameters. Following [11], we have the influence function defined as: $\mathcal{I}_{\text{up,params}}(\mathbf{x}_i) := \frac{d\hat{\theta}_{\epsilon,\mathbf{x}_i}}{d\epsilon}\Big|_{\epsilon=0} = -H_{\hat{\theta}}^{-1}\nabla_\theta\ell(\mathbf{x}_i; \hat{\theta})$ where $H_{\hat{\theta}}$ is the positive definite Hessian matrix [26]. Next, we can estimate the model parameters after adding this new sample $\mathbf{x}_i$, as: $\hat{\theta}_{\mathbf{x}_i} - \hat{\theta} \approx \frac{1}{|\mathcal{L}_{t-1}|+1}\mathcal{I}_{\text{up,params}}(\mathbf{x}_i) = -\frac{1}{|\mathcal{L}_{t-1}|+1}H_{\hat{\theta}}^{-1}\nabla_\theta\ell(\mathbf{x}_i; \hat{\theta})$ where $\nabla_\theta\ell(\mathbf{x}_i; \hat{\theta})$ could be approximated as the expected gradient of sample $\mathbf{x}_i$: By a slight abuse of notation of the training loss function $\ell$, we denote $\nabla_\theta\ell(\mathbf{x}_i; \hat{\theta}) \approx \sum_{k=1}^K \nabla_\theta\ell(\mathbf{x}_i, \hat{y}_k; \hat{\theta}) \cdot \hat{p}_k$ where $\hat{y}_k$ and $\hat{p}_k$ represent model's label prediction and likelihood (e.g. confidence) respectively while $K$ represents the total number of classes in the ground truths.

In practice, the inverse of $H_{\hat{\theta}}$ cannot be computed due to its prohibitive $O(np^2 + p^3)$ runtime [34], with $p$ being the number of model parameters. The computation unavoidably becomes especially intensive when $f$ is a deep neural network model [15]. Luckily, we have two optimization methods, conjugate gradients (CG) [37] and stochastic estimation [2], which are detailed in Appendix C.1.

### 2.4 Efficient Unbiased Loss Estimation

Referring back to Eq. (1), the last challenge that we need to address is to gain access to the unbiased true loss estimator $L_{\text{true}}$. In other words, we want to predict the *true* performance of $f_{t|\mathbf{x}_i,y_i}$ on the unlabeled set $\mathcal{U}_{t,i}$ without exhaustive testing. Strictly, such evaluation cannot be drawn until $f_{t|\mathbf{x}_i,y_i}$ is evaluated on the entire unlabeled set $\mathcal{U}_{t,i}$. However, this is infeasible in practice. Such approximation is typically carried out in other approaches [24, 41] by randomly sampling a labeled validation set $\mathcal{V}$ at the beginning of the entire acquisition process, which will later be used for evaluations in all the subsequent acquisition episodes. Despite the simplicity as well as being i.i.d., which makes the estimated loss unbiased by nature, this approximation method suffers from large variance as the size of $\mathcal{V}$ is usually much smaller than $\mathcal{U}$, which unavoidably hurts the acquisition performance. It is also contradictory to the goal of AL in general, especially under the low-budget settings.

Different from the existing works, we propose to sample a subset $\mathcal{C}$ from current $\mathcal{L}_{t-1}$ in each acquisition round based on an alternative acquisition function, and then correct the bias in the loss

induced from this acquisition function. In the meantime, we also want to keep the variance low, so that the final corrected loss enjoys both low bias and low variance, which is more preferable than the zero bias but high variance that the random i.i.d. sampling has. Specifically, continuing with the notations from §2.1, let $\mathcal{C} = \{\mathbf{x}_{t,1}, \ldots, \mathbf{x}_{t,m}, \ldots, \mathbf{x}_{t,n_{\mathcal{C}}}\}$, where $\mathcal{C} \subset \mathcal{U}_{t-1}$, be the subset containing $n_{\mathcal{C}}$ samples selected for this true loss estimation at each round $t$. [14] shows that if $\mathbf{x}_{t,m}$ is sampled in proportion to the true loss of each data point, the bias originated from this selection can be corrected through the Monte Carlo estimator $\hat{R}_{\text{LURE}}$[1]. Following our notations, it takes the form: $\hat{R}_{\text{LURE}} = \frac{1}{n_{\mathcal{C}}} \sum_{m=1}^{n_{\mathcal{C}}} v_m \ell(\mathbf{x}_{t,m}; f)$ where recall that $\ell$ denotes the loss of $f$, and the importance weight $v_m$ is

$$v_m = 1 + \frac{|\mathcal{U}_{t-1}| - n_{\mathcal{C}}}{|\mathcal{U}_{t-1}| - m} \left( \frac{1}{(|\mathcal{U}_{t-1}| - m + 1)q_t^*(m)} - 1 \right) \tag{4}$$

with $q_t^*(m)$ being the acquisition distribution of index $m$ at round $t$. Importantly, the variance can be significantly reduced if the acquisition distribution $q_t^*(m)$ is proportion to the true loss of each data point. Again, this is not feasible as we do not have access to the labels for $\mathcal{U}_{t-1}$. However, following [27], we can approximate $q_t^*(m)$ with $q_t(m) = -\sum_y \pi(y|\mathbf{x}_{t,m}) \log f(\mathbf{x}_{t,m})$ for classification tasks when the loss function is the cross-entropy loss; here $\pi$ is conveniently just our surrogate discussed in §2.2. Referring back to the discussion we had on choosing a good surrogate $\pi$,

---

**Algorithm 1** Direct Acquisition Optimization (DAO)

**input** Episode $t$, unlabeled set $\mathcal{U}_{t-1}$, labeled set $\mathcal{L}_{t-1}$, model $f_{t-1}$,
    surrogate $\pi_{t-1}$, budget $k$, $n_{\text{ihvp}}$ (§2.3), and $n_{\mathcal{C}}$ (§2.4)
**output** Acquisition set $\mathcal{A}_t = \{\mathbf{x}_{t,1}^{\text{train}}, \ldots, \mathbf{x}_{t,k}^{\text{train}}\}$         ▷ Eq. (1)
1: Approximate $p(y|\mathbf{x})$ for all $\mathbf{x} \in \mathcal{U}_{t-1}$ ▷ Appendix 2.2, Eq. (2)
2: Initialize array $S$ where $|S| = |\mathcal{U}_{t-1}|$
3: **for** $i = 1$ **to** $|\mathcal{U}_{t-1}|$ **do**
4:     Let $\mathcal{U}_{t,i} = \mathcal{U}_{t-1} \setminus \{\mathbf{x}_i\}$
5:     Randomly sample $n_{\text{ihvp}}$ data points from $\mathcal{U}_{t,i}$
6:     Approximate parameters of $f_{t|\mathbf{x}_i, y_i}$     ▷ §2.3, Eq. (6)
7:     Acquire $n_c$ samples from $\mathcal{U}_{t,i}$       ▷ §2.4, Eq. (5)
8:     Compute $s_i$ and add to $S$             ▷ §2.4
9: **end for**
10: Sort $S$ in ascending order
11: **if** $k > 1$ **then**
12:     Perturb $S$        ▷ Methods shown in Appendix C.2
13: **end if**
14: Return top-$k$ samples in $S$ as $\mathcal{A}_t$

---

with $f(\mathbf{x})$ being designed to approximate $p(y|\mathbf{x})$ as well, the surrogate $\pi$ should ideally be different from $f$ so that more diversity is introduced in the acquisitions. To put all components together, our loss correction process involves selecting samples in $\mathcal{C}$ following

$$\mathbf{x}_{t,m} \propto -\sum_y \pi_{t-1}(y|\mathbf{x}) \log f_{t-1}(y|\mathbf{x}) \tag{5}$$

where $\pi_{t-1}$ is the surrogate model at round $t - 1$. Finally, the corrected loss $s_i$ can be approximated using $\hat{R}_{\text{LURE}}$ as $s_i = \frac{1}{n_{\mathcal{C}}} \sum_{m=1}^{n_{\mathcal{C}}} \hat{v}_m \ell(\mathbf{x}_{t,m}; f_t)$ where $\hat{v}_m$, which depends on the choice of $\mathbf{x}_{t,m}$, is the approximated version of the original $v_m$ defined in Eq. (4). Specifically, $\hat{v}_m$ takes the form $\hat{v}_m = 1 + \frac{|\mathcal{U}_{t-1}| - n_{\mathcal{C}}}{|\mathcal{U}_{t-1}| - m} \left( \frac{1}{(|\mathcal{U}_{t-1}| - m + 1)q_t(m)} - 1 \right)$, where $q_t(m)$ is the acquisition function in Eq. (5). we summarize the components illustrated in §2 and present it in Algorithm 1.

## 3 Experiments

### 3.1 Experiments Setup

**Baselines.** To ensure fair comparisons, besides baseline methods that we empirically surveyed in Appendix A, we also include other state-of-the-arts AL methods, including Deep Bayesian Active Learning (DBAL) [16] and GLISTER [24], where GLISTER is a direct competitor that also optimizes the EER framework. For all the baselines, we used the default/recommended parameters and their official implementations if publically available. In terms of earlier works such as least confidence [31], minimum margin [45], and maximum entropy [49], we used the peer-reviewed deep active learning framework DeepAL+ [59]. All experiments are repeated ten times with different random seeds.

**Implementation Details.** We use ResNet-18 [19] as the primary model $f$, trained from scratch, and VGG16 [51] with randomly initialized weights as the surrogate model $\pi$. For estimating updated

---

[1]LURE stands for Levelled Unbiased Risk Estimator

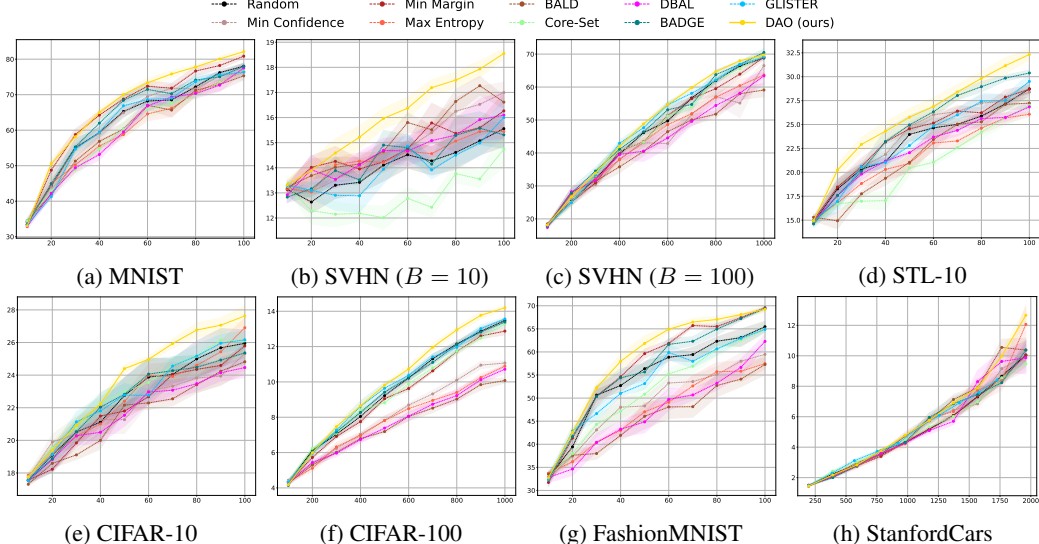

(a) MNIST     (b) SVHN ($B = 10$)     (c) SVHN ($B = 100$)     (d) STL-10

(e) CIFAR-10     (f) CIFAR-100     (g) FashionMNIST     (h) StanfordCars

Figure 2: Experiment results showing DAO compared to other AL algorithms across seven benchmarks, with labeled set size on the horizontal axis and classification accuracy on the vertical axis.

model parameters, we apply stochastic estimation [2], as detailed in §2.3. We set $n_{\text{ihvp}} = 8$ when approximating the unbiased estimator of $H_{\hat{\theta}}$ and $n_{\mathcal{C}} = 16$ for biased loss correction, following §2.4.

### 3.2 Results

We evaluate DAO on two digit recognition benchmarks: MNIST [30], a dataset of 60k handwritten digit images, and SVHN [48], a more challenging dataset with over 600k street-view house number images. Both datasets have 10 classes (digits 0-9). In low-budget settings, we use one image per class, resulting in an initial label size of 10 and a budget of 10 per round for MNIST. For the larger SVHN dataset, we experiment with initial labeled sizes and budgets of 10 and 100. Results are shown in Fig. 2b and Fig. 2c. We further test DAO on more complex object classification tasks using STL-10 [10], CIFAR-10, and CIFAR-100 [29]. STL-10 has 5k labeled 96x96 color images across 10 classes, with 8k test images. CIFAR-10 contains 60k 32x32 images in 10 classes, while CIFAR-100 expands to 100 classes with 600 images per class. In the low-budget setting (1 image per class), we use initial label sizes and per-round budgets of 10 for STL-10 and CIFAR-10, and 100 for CIFAR-100. Results are displayed in Fig. 2d, Fig. 2e, and Fig. 2f. The final part of our experiments focuses on applying DAO to domain-specific tasks. We use FashionMNIST [57] and StanfordCars (Cars196) [28]. FashionMNIST, similar in structure to MNIST, consists of 70k 28x28 images of fashion products from 10 categories, with 60k images for training and 10k for testing. StanfordCars contains 16,185 images of cars, split into 8,144 for training and 8,041 for testing, across 196 classes representing car make, model, and year (e.g., 2012 Tesla Model S). Results are in Fig. 2g and 2h.

### 3.3 Discussion

From Fig. 2, we observe that DAO consistently outperforms state-of-the-art active learning methods across all seven benchmarks. Notably, in the SVHN dataset with an extremely low budget ($B = 10$, 0.0017% of the unlabeled set), DAO demonstrates a significant performance advantage, highlighting its strength in low-budget settings. As the budget increases, DAO continues to perform well, as seen in Fig. 2c. The only dataset where DAO shows less improvement is StanfordCars. However, DAO still provides smoother accuracy gains with less variance, indicating greater robustness in complex tasks like StanfordCars, which has 196 classes.

## 4 Conclusions

In this paper, we introduced Direct Acquisition Optimization (DAO), a novel algorithm designed to optimize sample selections in low-budget settings. DAO hinges on the utilization of influence functions for model parameter updates and a separate acquisition strategy to mitigate bias in loss estimation, represents a significant optimization of the EER method and its modern follow-ups. Through empirical studies, DAO has demonstrated superior performance in low-budget settings, outperforming existing state-of-the-art methods by a significant margin across seven datasets.

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

# Appendix

## A  Low-Budget Active Learning: A Motivating Case Study

In this section, we provide an empirical analysis to demonstrate that commonly used heuristic-based AL algorithms do not work well under very low-budget settings. Specifically, we analyze (1) uncertainty sampling methods including least confidence [31], minimum margin [45], maximum entropy [49], and Bayesian Active Learning by Disagreement (BALD) [16]; (2) diversity sampling methods such as Core-Set [47] and Variational Adversarial Active Learning (VAAL) [52]; and (3) hybrid method such as Batch Active learning by Diverse Gradient Embeddings (BADGE) [4].

We test the above methods on the CIFAR10 [29] dataset starting with an initial labeled set with size $|\mathcal{L}_{\mathrm{init}}| = 10$, and conducted 50 acquisition rounds where after each round $B = 10$ new samples are selected and labeled. We use ResNet-18 [19] as our training model across all methods. And we repeated the acquisitions five times with different random seeds. The results are visualized in Fig. 3, where we plot the *relative* performance between each method and random sampling acquisition through a diverging color map.

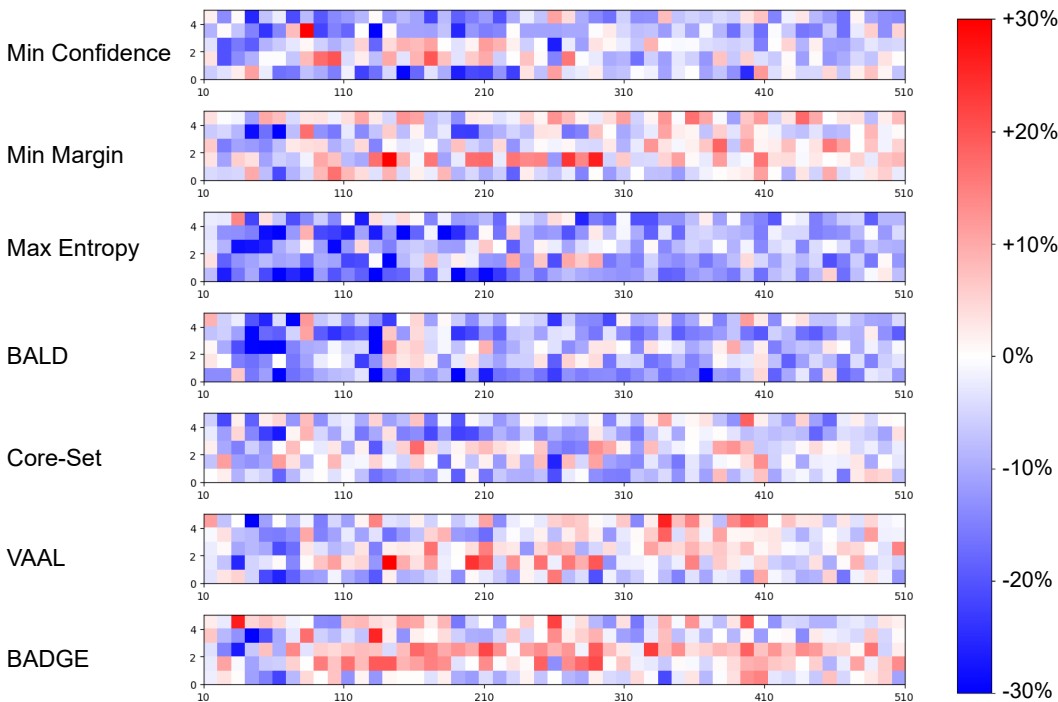

Figure 3: Existing methods fail to outperform random sampling with small budgets. This figure shows the relative performance between multiple methods and random acquisition. Within each subplot, $x$ axis represents the accumulative acquisition size, while $y$ axis indicates runs initiated with different random seeds. White color indicates on-par performance with random, blue indicates worse, and red indicates better.

Aligning with the general perceptions that low-budget [39, 18] and cold-start [60, 8] AL tasks are especially challenging, we empirically observe that almost all popular AL algorithms fail to outperform the naive random sampling when acquisition quota is less than 1% (500 out of 50,000 in the case of CIFAR10) of the unlabeled size. More specifically, when the quota is less than 0.2% (less than 100 data points for CIFAR-10), all methods fail to reliably outperform random sampling (as the beginning of each heatmap in Fig. 3 are almost all blue), which greatly motivates the development of DAO. We also include the more conventional line plot of the empirical analysis which may provide more detailed information of each run in Fig. 4.

## B  Related Work

**Active learning.** AL has gained a lot of attraction in recent years, with its goal to achieve better model performance with fewer training data [49, 46, 42]. There have been different selection

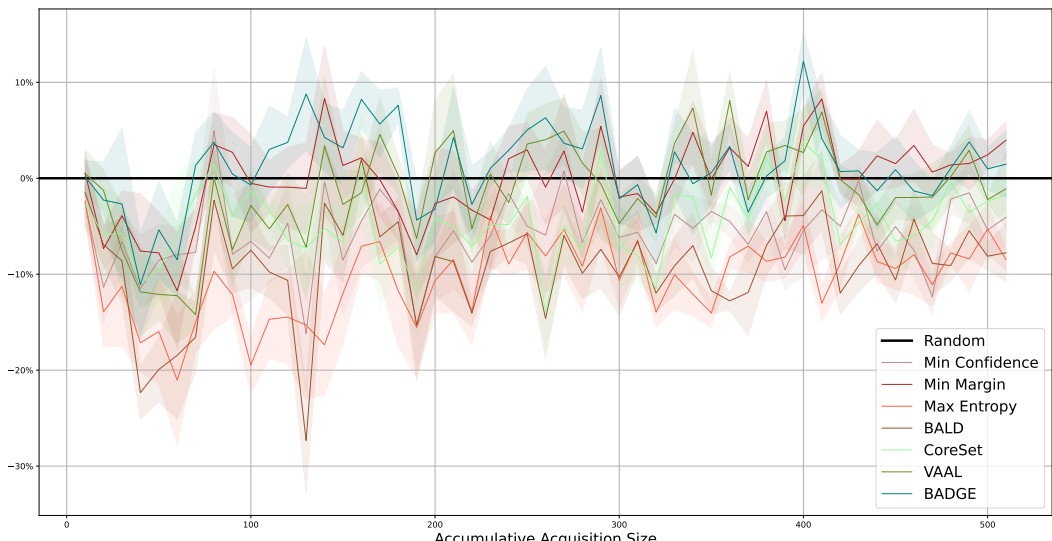

Figure 4: Relative performance between existing popular AL methods and random acquisition. horizontal axis represents the accumulative size of the labeled set, while vertical axis indicates relative performance in percentage.

criteria including uncertainty, diversity, query-by-committee, version space and information-theoretic heuristics [33, 59]. The uncertainty-based approaches are arguably the most popular and easiest to implement, which includes selection criteria such as least confidence [31], minimum margin [45, 43, 9], maximum entropy [23, 49] and others [16]. At their core, these methods select points where the classifier is least certain. However, uncertainty-based methods can be biased towards the current learner. Diversity-based methods [49, 6, 17, 35, 13, 36, 58, 47, 52, 3, 56], on the other hand, aim to select the most representative samples of the dataset. In addition, query-by-committee [50, 1] and version space-based [38] methods, keep a pool of models, and then select samples that maximize the disagreements between them. Information-theoretic methods [20, 5] typically utilize mutual information as the criterion. Hybrid method that combines both uncertainty and diversity criteria, such as BADGE [4], has also been developed to take advantage of both worlds. As shown later in the paper, we visually observe that the selections of our proposed DAO, although not explicitly optimized towards any of these heuristics, display characteristics of an hybrid approach.

**EER-based acquisition criterion.** Alternatively, EER was proposed to select new training examples that result in the lowest expected error on future test examples, which directly optimizes the metric by which the model will be evaluated [44]. In essence, EER employs sample selection based on the estimated impact of adding a new data point to the training set, rather than evaluating performance against a separate validation set, meaning that it does not inherently require a validation hold-out set. However, its necessity to retrain the model for every possible candidate sample and every possible label renders its cost intractable in the context of deep neural networks [7, 53, 12]. More recent look-ahead EER-based AL algorithms [41] focus on addressing this efficiency concern. However, these methods either rely on a small set of validation data to be used for the evaluation of the expected loss reduction [24], or can still be quite slow when the size of labeled and unlabeled sets are large [40]. In this paper, we present DAO, a novel AL algorithm that improves upon EER through optimizations on both model updates as well as loss estimation, efficiently and effectively broadening the applicability of EER-based algorithm.

# C  More Detailed Methodology

## C.1  Computational Optimization in Model Parameter Update

**Conjugate gradients.** As mentioned earlier, by assumption we have $H_{\hat{\theta}} \succ 0$ and $\nabla_\theta \ell(\mathbf{x}'; \hat{\theta})$ as a vector. Therefore, we can calculate the inverse Hessian vector product (IHVP) through first transforming the matrix inverse into an optimization problem, i.e.

$$H_{\hat{\theta}}^{-1} \nabla_\theta \ell(\mathbf{x}_i; \hat{\theta}) \equiv \arg \min_t \; t^T H_{\hat{\theta}} t - v^T t$$

and then solving it with CG [37], which speeds up the runtime effectively to $O(np)$.

**Stochastic estimation.** Besides CG, we can also efficiently compute the IHVP using the stochastic estimation algorithm developed by Agarwal et al. [2]. From Neumann series, we have $A^{-1} \approx \sum_{i=0}^{\infty} (I - A)^i$ for any matrix $A$. Similarly, suppose we define the first $j$ terms in the Taylor expansion of $H_{\hat{\theta}}^{-1}$ as

$$H_{\hat{\theta},j}^{-1} = \sum_{i=0}^{j} (I - H_{\hat{\theta}})^i = I + (I - H_{\hat{\theta}}) H_{\hat{\theta},j-1}^{-1}$$

we have $H_{\hat{\theta},j}^{-1} \to H_{\hat{\theta}}^{-1}$ as $j \to \infty$. The core idea of the stochastic estimation is that the Hessian matrix $H_{\hat{\theta}}$ can be substituted with any unbiased estimation when computing $H_{\hat{\theta}}^{-1}$. In practice, we sample $n_{\text{ihvp}}$ data points from the existing labeled set $\mathcal{L}_{t-1}$ and use $\nabla_\theta^2 \ell(\mathbf{x}_i; \hat{\theta})$ as the estimator of $H_{\hat{\theta}}$ [34]. Notice that since $n_{\text{ihvp}}$ is usually very small (in our experiments we used $n_{\text{ihvp}} = 8$), it does not create a constraint on the size of the current labeled set, which does not interfere with the low-budget settings. Finally, we can approximate the model parameters after the addition of $\mathbf{x}_i$ as

$$\hat{\theta}_{\mathbf{x}_i} = \hat{\theta} - \frac{1}{n+1} H_{\hat{\theta}}^{-1} \nabla_\theta \ell(\mathbf{x}_i; \hat{\theta}) \tag{6}$$

which does not require any re-training. And we will demonstrate in §E.1 that this parameter update strategy provides much better approximations than the naive single backpropagation as seen in the existing AL literature [24].

## C.2  Batch Acquisition via Stochastic Sampling

In §2.1, we briefly discussed that when $\mathbf{x}_t^{\text{train}}$ represents a set of data points (instead of a single one), the formulation in Eq. (1) essentially represents the *batch* active learning scenario. Suppose the acquisition budget per round is $k$, although selecting the top $k$ samples with the lowest estimated losses (or highest expected error reduction) is straightforward, this approach is sub-optimal. This is because top-$k$ acquisition, while effective to some degree due to its greedy nature, overlooks the crucial interactions among data points in batch acquisitions. Specifically, while aiming to select the most informative unlabeled points, top-$k$ acquisition may lead to redundant choices, diminishing the overall benefit of the acquisition.

Inspired by **(author?)** [25], we propose to similarly perturb the original ranking of the estimated true losses so that the batch sampling provides better acquisitions when the most informative data points may be duplicated. Suppose at acquisition episode $t$, we rank the set of estimated true loss of each unlabeled data point in ascending orders as $\{\hat{l}_{\text{true},i}\}_{\mathbf{x}_i \in \mathcal{U}_{t-1}}$, such that $\hat{l}_{\text{true},i} \le \hat{l}_{\text{true},j}, \forall i \le j$ and $\mathbf{x}_i, \mathbf{x}_j \in \mathcal{U}_{t-1}$, we can perturb the ranking with three strategies: soft-rank, soft-max, and power acquisition, to improve batch performance from the naive top-$k$ sampling.

**Soft-rank acquisition.** Soft-rank acquisition relies on the relative ordering of the scores while ignoring the absolute score values. It samples the data point ranked at index $i$ with probability $p_{\text{softrank}}(i) = i^{-\beta}$, where $\beta$ is the "coldness" parameter and is kept as 1 throughout this paper. It is not hard to notice that $p_{\text{softrank}}(i)$ is invariant to $\hat{l}_{\text{true},i}$, as long as the relative ranking remains the same. More conveniently, with sampled Gumbel noise $\epsilon_i \sim \text{Gumbel}(0; \beta^{-1})$, taking the top-$k$ data points from the perturbed ranked list

$$\hat{l}_{\text{true},i}^{\text{softrank}} = -\log i + \epsilon_i$$

is equivalent to sampling $p_{\text{softrank}}(i)$ without replacement [22].

**Soft-max acquisition.** In contrast to soft-rank, soft-max acquisition uses the actual scores, i.e., the estimated true losses, instead of their relative orderings. However, this acquisition does not rely on the semantics of the actual values, resulting in the transformed true loss simply being:

$$\hat{l}_{\text{true},i}^{\text{softmax}} = \hat{l}_{\text{true},i} + \epsilon_i$$

where $\epsilon_i$ remains the same Gumbel noise as in the soft-rank acquisition. Statistically, choosing the top-$k$ data points from this perturbed ranked list is equivalent to sample from $p_{\text{softmax}}(i) = e^{\beta i}$ without replacement.

**Power acquisition.** While neither soft-rank or soft-max acquisitions take the semantic meaning of the actual score values into account when designing the acquisition distribution, power acquisition uses the value directly when determining the perturbed values. Specifically, the power acquisition perturbs the scores as

$$\hat{l}_{\text{true},i}^{\text{power}} = \log \hat{l}_{\text{true},i} + \epsilon_i$$

where again $\epsilon_i$ is the Gumbel noise, and choosing the top-$k$ indices from this new list is equivalent to sampling from $p_{\text{power}}(i) = i^{\beta}$ without replacement. Results comparing DAO with different batch acquisition strategies discussed above are showed in Appendix E.5. Combining all the components, the pseudocode of DAO is summarized in Algorithm 1.

# D   Visualizations of Samples Selected by DAO

In this section, we show the visual representations of the data samples collected by DAO as a complement to §E.4. Unlabeled and newly acquired data, in this case, images, or their latent space embeddings, are first dimensionally-reduced and then visualized in Fig. 5. We see that, DAO-selected data exhibit characteristics of diversity across the sample space over multiple acquisition rounds, while display uncertainty characteristics within single round.

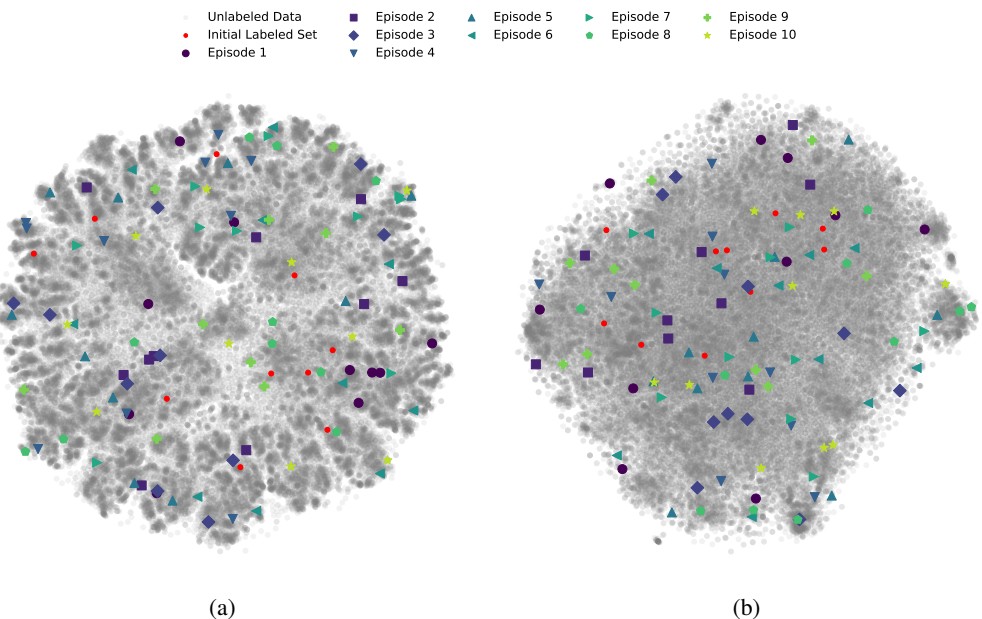

(a)                                              (b)

Figure 5: Visualizations of DAO acquisitions with dimensionality reduced from (a): raw images; and (b): latent space image embeddings.

# E Component Analysis and Ablation Studies

## E.1 Accuracy on Model Approximation

First, we assess if estimating the model parameters updates through modelling the effect of adding a new sample as upweighting the influence function provides a more accurate model performance approximation than using single backpropagation as seen in GLISTER [24]. Specifically, we conduct the experiments on CIFAR-10 [29], with initial labeled size $|\mathcal{L}_{\text{init}}^{\text{CIFAR-10}}| = 100$ (randomly sampled from the train split), per-episode budget $B_{\text{CIFAR-10}} = 1$, and number of acquisition episode $E = 25$. We compare the updated models performance (accuracy) on the test split of CIFAR-10. Different from the experiments in §3, we do not apply any AL algorithm when acquiring the sample in each round.

Instead, we randomly choose $B$ sample in each acquisition round from the unlabeled set and then update the models through both methods with the same selected sample.

To access the difference between models updated with our influence function-based method and single backpropagation, we compute the mean squared error (MSE) between the performance of each model and the model updated by conventional full training, which is defined in Eq. (3). As shown in Fig. 6a, the proposed method provides more accurate (smaller mean and median) and more robust (smaller std.) model approximations than single backpropagation, contributing to the performance gain we observe in §3.

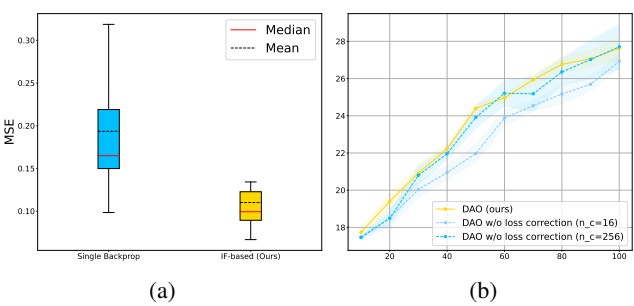

(a)            (b)

Figure 6: *Left*: MSE of the predictions accuracy on the test split of CIFAR-10 between models updated by single backpropagation, influence function, and the fully trained model. *Right*: Ablation results where the proposed loss estimation is replaced by the random sampling estimation defined in §E.2.

## E.2 Bias Correction vs. Random Sampling

Next, we conduct ablation studies on replacing the proposed loss estimation (§2.4) with the average loss of randomly sampled data points. More specifically, we replace the estimated loss $s_i$ from averaging the corrected loss of the acquired samples via an alternative acquisition criteria (Eq. (5)) with averaging losses of the samples acquired uniformly, i.e., at round $t$, we have $s_i^{\text{random}} = \frac{1}{M_{\text{random}}} \sum_{m=1}^{M_{\text{random}}} \ell(\mathbf{x}_{t,m}; f_t)$ where $\mathbf{x}_{t,m} \sim U(1, |\mathcal{U}_{t,i}|)$. We choose two $M_{\text{random}} = 16$ and 256, where former provides a direct comparison with our proposed loss estimation approach, and latter represents a brute-force solution that works relatively well but is often infeasible in practice due to intensive running time. The results are shown in Fig. 6b. We see that the proposed method performs even better than the conventional random-sampling loss estimation with large sampling size, while computationally being only 1/8 of the run time. Additionally, the variance of our method is much smaller, indicating more robust loss estimation and thus more robust acquisition performance.

## E.3 Different Batch Acquisition Strategies

We conducted additional ablation studies comparing various stochastic sampling methods as detailed in Appendix C.2. Results are documented in Appendix E.5. Our findings reveal that the proposed algorithm, even when simply selecting the top $k$ samples without applying any of the stochastic strategies, outperforms existing methods. Performance further improves with the implementation of these sampling strategies. It is important to note that we have not designed specific sampling strategies for our algorithm; instead, we utilized existing methods to showcase the efficacy of DAO framework.

## E.4 Interpreting DAO with Other AL Criteria

In this section, we analyze the criterion optimized by DAO and compare it to common criteria such as diversity and uncertainty, using visual representations of the data samples collected by DAO. The detailed plots are available in Appendix D. Throughout multiple acquisition rounds, the data selected by DAO demonstrate notable diversity with uniform distribution across the sample space. However, in contrast to traditional uncertainty-based methods, selections within a single round by DAO also

incorporate elements of uncertainty. This hybrid approach explains the performance improvements observed in §3 over algorithms that solely focus on diversity or uncertainty.

## E.5 Experiments on Different Stochastic Batch Acquisitions

In this section, we provide more detailed results of §E.3. Specifically, we further study the performance of DAO when no batch sampling strategy or other sampling strategy is used and compare the results with existing popular AL algorithms. The results are shown in Fig. 7. For all experiments, we used the same low-budget setting as discussed in §3.

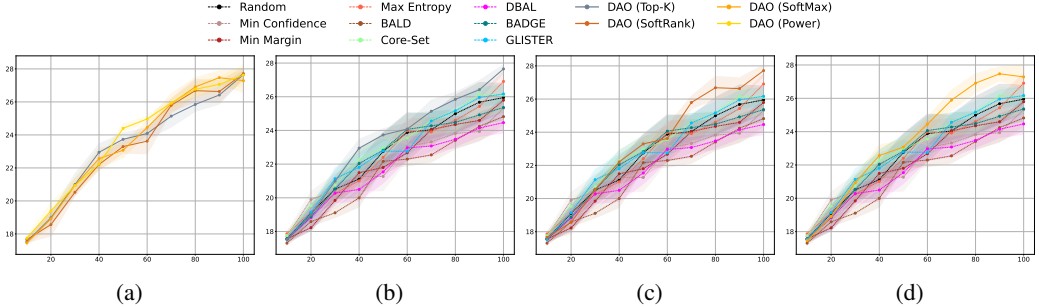

Figure 7: CIFAR-10 experiment results on (a): DAO without batch acquisition strategy (using naive top-k selection) and with other sampling strategies (softmax and softrank, as discussed in §C.2); (b): DAO without sampling (top-k) vs. existing AL algorithms; (c): DAO with softrank sampling vs. existing AL algorithms; (d): DAO with softmax sampling vs. existing AL algorithms; In all subplots, horizontal axis represents the accumulative size of the labeled set, while vertical axis indicates classification accuracy.

## E.6 Ablations on Different Surrogates

To further clarify how the surrogate model impacts the performance of DAO, we conducted additional experiments on CIFAR-10 using different surrogates. Specifically, we compared: (1) *VGG16*: The surrogate model currently used in the paper draft. (2) *ResNet18*: An efficient variant of DAO where the main model under training serves as the surrogate, i.e., $\pi_t = f_t$. In this case, the equation for approximating $q_t^*(m)$, the acquisition distribution of candidate index $m$ at round $t$ (the equation between lines 225 and 226), reduces from cross-entropy to entropy. (3) *SimpleCNN*: A simpler version of ResNet18 with six convolution blocks, each containing one Conv layer followed by BatchNorm and ReLU. and (4) *Oracle*: An unrealistic setting assuming access to an oracle surrogate model. We used the same experimental setup with CIFAR-10 as in our draft: starting with 10 labeled samples, acquiring 10 samples per round, and continuing for 10 rounds. The was repeated five times with different random seeds.

As shown in Fig. 8, DAO with oracle surrogate performs the best, followed by VGG16 and SimpleCNN. ResNet18 performs the worst among all DAO variants, which aligns with our expectation that performance degrades when $\pi_t$ never disagrees with $f_t$. However, all DAO variants outperform the random and GLISTER baselines by a clear margin. It is worth noting that while the oracle surrogate achieves the best results, the improvement over VGG16 and SimpleCNN is not substantial. We think this is likely because, when selecting samples for the unbiased loss estimation, the acquisition distribution $q_t^*(m)$ approximated with $q_t(m) = -\sum_y \pi(y|\mathbf{x}_{t,m}) \log f(\mathbf{x}_{t,m})$, does not solely depend on the surrogate quality. Although in

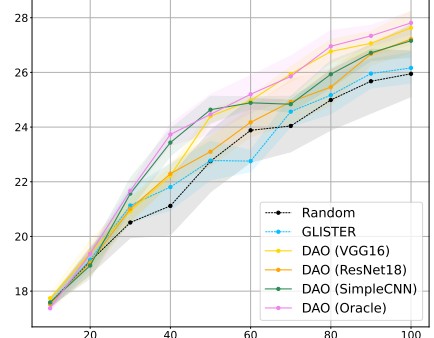

Figure 8: Exp. with different surrogates.

the unrealistic case of an oracle surrogate, this creates an disadvantage, but in practical scenarios, this approximation provides better robustness in preventing the negative impact of poor quality of surrogate on unbiased loss estimation, especially in early stages where models might overfit.

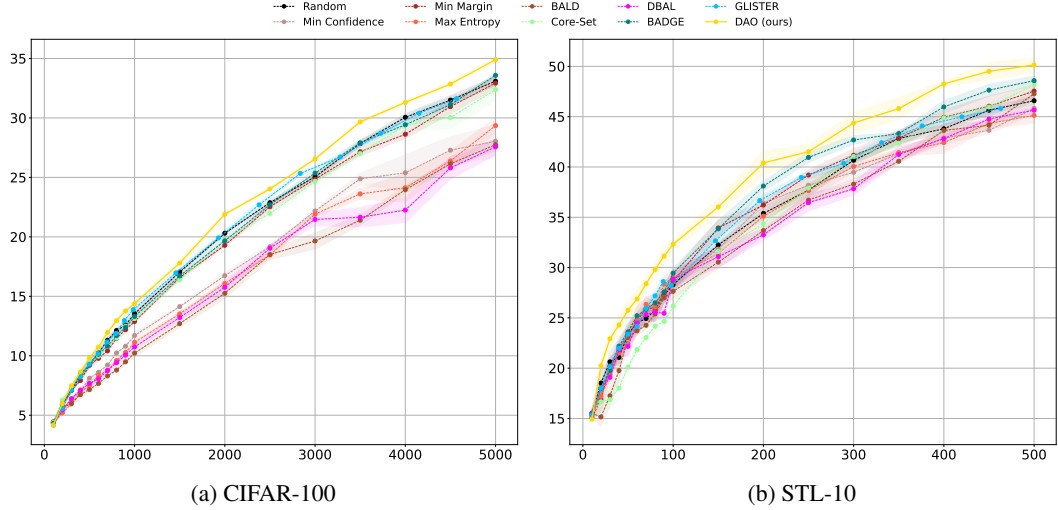

(a) CIFAR-100           (b) STL-10

Figure 9: Higher-budget experiment results comparing DAO with existing AL algorithms. In both subplots, horizontal axis represents the accumulative size of the labeled set, while vertical axis indicates classification accuracy.

# F    Experiments with Higher Total Budgets

To further evaluate the capabilities of the proposed DAO beyond its initial focus on low-budget active learning, we conducted additional experiments with higher budgets on the CIFAR-100 and STL10 datasets. Specifically, we used the same experimental settings as in the draft, with the same initial labeled set and per-round acquisition budget, and repeated the process five times with different random seeds. However, we extended the number of rounds in each acquisition from 10 to 50, increasing the budget by five times. To make the plot more clear, we plot every five rounds, and the results are shown in Fig. 9.