# OpenReview forum: "Direct Acquisition Optimization for Low-Budget Active Learning"
_NeurIPS.cc/2024/Workshop/BDU — NeurIPS BDU Workshop 2024 Poster_

### Official Review · Reviewer_6pCL · 2024-09-24
**Review of DAO**

**Rating:** 7
**Confidence:** 4

**Review:**

The article Direct Acquisition Optimization (DAO) for Low-Budget Active Learning offers insightful and innovative research proposals in the field of active learning. In general, its approach utilizing  influence functions to update the model parameter, as opposed to retraining the predictor model at each iteration, is innovative and clever. DAO has the potential to be advantageous in overcoming common problems in the EER-methods literature. In particular, it addresses two issues in EER methods: (i) the need to hold out a validation dataset and (ii) the computational expense of retraining the model for each candidate observation.

DAO's contribution in (i) addresses the fundamental motivation of active learning: a study should utilize its labelling-budget, often extremely constrained, on labelling the most informative observation. As such, labelling and preserving a validation set not used in the training of the predictive model may be wasteful. DAO cleverly overcomes this by its non-need to retrain and revalidate. Note that reading Appendix C was needed to fully understand why this works in their methodology. The author's methods builds on the idea of surrogates, an idea from the related active testing literature. I am not entirely familiar with the active testing literature - it is similar yet different to the active learning literature - but their use of surrogates overcomes the issue of not knowing the true label distribution with a good approximation. It would make the author's methods more clear if their use of surrogates is further explained.

DAO's needlessness to retrain the model on each candidate observation overcomes issue (ii). As EER methods have not been commonplace in deep neural networks due to the large computational cost associated with this training, DAO's circumventing of this expensive retraining has the potential contribute to the active learning literature at large. Section C.3 and E on computational optimization well explains why their parameter update strategy approximates the model parameters well.

Overall, the article outlines their research proposal very well and provides novel methodological improvements to the active learning literature. The following comments below outline additional sections that I found unclear.

In section 2.1, $\hat{\theta}_{x_i}$ is defined in equation (1) as the parameter to which the training model converges to, but

$\hat{\theta}_{\epsilon, x_i}$
should be specified in the influence function.

In section 2.2, the authors define a candidate subset $\mathcal{C}$ for oracle labelling as being within the labelled dataset $\mathcal{L}$. However, they later define $\mathcal{C} \subseteq \mathcal{U}$.

The axes of Figure (2) are unclear.

I see from the experimental results of Section 3 that DAO indeed provides a predictive advantage over the other models with less predictive variance. It would be interesting to see the theoretical foundations behind this result, or if this was just the case in the experiments. Likewise, Section E.4 show that DAO incorporates elements of diversity in its acquisition procedure. The connection is clear from the image in Section D but the reasoning behind this finding is not clear. A formalization of how/why DAO integrates acquisition diversity would strengthen the author's promotion of DAO.

---

### Official Review · Reviewer_WV53 · 2024-09-26
**Interesting active learning algorithm and impressive experimental results**

**Rating:** 8
**Confidence:** 2

**Review:**

This paper presents Direct Acquisition Optimization (DAO), a novel active learning algorithm based on expected error reduction (rather than heuristics such as example diversity or model uncertainty).  DAO employs influence functions to approximate the effect of training on candidate unlabeled data, and DAO samples from the unlabeled data for loss estimation and then applies bias correction via LURE (instead of setting aside a validation set beforehand).

The results are impressive, showing DAO out-performing many baseline algorithms including BALD, BADGE, DBAL, and GLISTER on seven benchmark datasets in low-budget settings with a small number of rounds of data acquisition.

The paper is well written, and the topic of active learning is explicitly included in the theme for this workshop.  So, this paper should be accepted.

Nit-picks:
1. Line 60: Is the reference to the appendix intentional?
2. Line 66: "could has been" -> "could have been"
3. Line 70: Delete "recall that"
4. Line 86: This line refers "back" to an equation that hasn't been encountered yet and is in the appendix.  It feels like the paragraphs of the paper have been shuffled and some of the references make less sense now.  The subsequent paragraphs also refer to notation and discussion in appendix C.
5. Line 162: The second sentence in the conclusion should probably be split into two sentences.
6. Line 335: Delete the commas on this line.
7. Line 337: Consider rewriting this sentence as "Hybrid methods, such as BADGE [4], combine both uncertainty and diversity criteria."
8. Line 340: "an hybrid" -> "a hybrid"

---

### Decision · Program_Chairs · 2024-10-09

Accept (Poster)